# Impact of microRNA Regulated Macrophage Actions on Adipose Tissue Function in Obesity

**DOI:** 10.3390/cells11081336

**Published:** 2022-04-14

**Authors:** Alyssa Matz, Lili Qu, Keaton Karlinsey, Beiyan Zhou

**Affiliations:** 1Department of Immunology, School of Medicine, University of Connecticut, Farmington, CT 06030, USA; amatz@uchc.edu (A.M.); lili@uchc.edu (L.Q.); karlinsey@uchc.edu (K.K.); 2Institute for Systems Genomics, University of Connecticut, Farmington, CT 06030, USA

**Keywords:** obesity, immune response, macrophage, microRNA

## Abstract

Obesity-induced adipose tissue dysfunction is bolstered by chronic, low-grade inflammation and impairs systemic metabolic health. Adipose tissue macrophages (ATMs) perpetuate local inflammation but are crucial to adipose tissue homeostasis, exerting heterogeneous, niche-specific functions. Diversified macrophage actions are shaped through finely regulated factors, including microRNAs, which post-transcriptionally alter macrophage activation. Numerous studies have highlighted microRNAs’ importance to immune function and potential as inflammation-modulatory. This review summarizes current knowledge of regulatory networks governed by microRNAs in ATMs in white adipose tissue under obesity stress.

## 1. Introduction

The increasing prevalence of obesity is a global problem that burdens the health care system with severe co-morbidity and mortality [1,2,3]. White adipose tissue (WAT) dysfunction plays a central role through two primary contributing aspects: low-degree chronic inflammation and insulin resistance, as well as other health risk factors such as hyperlipidemia and hypertension [4]. Studies have demonstrated that controlling obesity-associated WAT inflammation can improve tissue function and systemic health outcomes [5,6,7,8,9]. Given the significance of WAT expansion and its pathogenic role in obesity, we will focus on summarizing the current knowledge of microRNA-regulated adipose tissue macrophage (ATM) actions that impact WAT function in this review. 

### 1.1. Macrophages Are Crucial to Adipose Tissue Homeostasis and Perpetuate Inflammation 

WAT inflammation activates and is reinforced by pro-inflammatory adipose tissue macrophages (ATMs), the most abundant immune populations in obese WAT [10,11]. In addition to their innate immune responses, ATMs also serve a vital role in maintaining AT function for proper tissue homeostasis via dead adipocyte clearance, regulating adipogenesis and angiogenesis, facilitating extracellular matrix (ECM) remodeling, and modulating lipid metabolism [12,13]. 

Macrophages are integral to the innate immune system. Macrophage plasticity allows rapid and diversified responses to complex endogenous and exogenous stimuli. Throughout the body, macrophages maintain homeostasis by exerting various functions tailored to each tissue. ATMs represent the most abundant immune population within AT. ATMs originate from hematopoietic stem-cell-derived circulating monocytes and self-replicating tissue residents that are seeded during fetal development [10,14,15]. During obesity, the ATM population increases 10-fold in cell number compared to their counterparts in lean WAT. Macrophages accumulate in obese WAT through monocyte-derived infiltration, tissue-resident expansion, and enhanced tissue retention. Although macrophage function is not dictated by their ontogeny [16,17], specifically inhibiting the infiltration of monocyte-derived macrophages into WAT during obesity lessens tissue inflammation [14,18,19]. 

Actions of ATMs are highly tissue-specific with a wide array of responses to a given physiological condition, such as obesity, and tightly controlled by a well-orchestrated network including microRNAs. Macrophage activation status can be described along a spectrum of pro-inflammatory M1 and the anti-inflammatory M2 phenotype, described below.

### 1.2. MicroRNA Biogenesis, Mechanism of Action, and Regulation of Macrophage Function

Heterogeneous macrophage functions are initiated by diverse, multilayered signals from their microenvironment and shaped by epigenetic regulatory elements such as microRNAs [20]. MicroRNAs are ~18–23 base pairs-long, non-coding RNAs that tune initiation, magnitude, and resolution of various cellular actions. 

#### 1.2.1. MicroRNA Biogenesis

Biogenesis of mature, functional microRNAs is controlled at multiple levels, including translation, nuclear processing, and cytoplasmic processing [21,22]. MicroRNAs biogenesis can follow either canonical or non-canonical pathways. However, most functional microRNAs are processed via the canonical pathway and thus will be described herein. For a complete review of canonical and non-canonical microRNA biogenesis, see [20,23]. 

*Transcription*: Canonical microRNAs are encoded in the genome as long primary transcripts (pri-miRNA). Most pri-miRNAs are transcribed by RNA polymerase II [24]. Many canonical microRNA genes are within intergenic non-coding and introns of coding regions and thus can be subject to the control of the same promoters of the host genes or the promoters of adjacent genes [25]. Pri-miRNAs contain a 33–35 base-paired stem-loop structure, in which a mature microRNA sequence and its complementary strand are embedded. Canonical microRNA biogenesis is dependent on the nuclear ribonuclease III Drosha. Drosha crops pri-miRNA at the stem-loop’s base to release the microRNA-containing stem-loop for further processing [26,27]. The released stem-loop is termed the pre-miRNA. Drosha relies on its co-factor DGCR8 to bind the double-stranded RNA (dsRNA) stem-loop of the pri-miRNA [26,27,28]. However, how the Drosha-DGCR8 complex localizes to the stem-loop base is unclear. It is known that the stem-loop’s sequence does not determine Drosha-DGCR8 binding; rather, the presence of an 80-nucleotide-long stem-loop structure accompanied by unpaired flanking sequences is necessary for the Drosha-DGCR8 complex to process pri-miRNAs [28,29]. 

*Maturation*: In canonical biogenesis, the stem-loop structured pre-miRNA is exported to the cytoplasm via Exportin-5 in a RAS-related nuclear protein-guanosine triphosphate (RAN-GTP)-dependent manner [30]. Drosha dsRNase activity produces a two-nucleotide-long 3′ overhang in the pre-miRNA [27], allowing the binding of cytoplasmic dsRNase Dicer [31]. Dicer interacts with its co-factor, TAR RNA-binding protein (TRBP) [32], to cut pre-miRNAs 21–25 nucleotide away from the 3′ terminus, or 22 nucleotides from the 5′ terminus [33,34], releasing the mature microRNA and its complement as a small RNA duplex.

#### 1.2.2. Formation of the RNA-Induced Silencing Complex (RISC)

In order for mature microRNAs to exert their function, they must be loaded onto an Argonaute (AGO) protein to form the RNA-induced silencing complex (RISC) [35]. The entire small RNA duplex is loaded into the AGO protein to form a pre-RISC. During the final maturation process, the RNA duplex then unwinds, and only one strand, the mature microRNA or “guide RNA”, is maintained in the functional RISC complex [36]. Whether the 5′ strand or the 3′ strand of the duplex becomes the guide strand can vary across tissues and activation status, and the different strands can target different mRNAs [37,38,39,40]. For example, miR-142-5p is the dominant isoform in the brain, ovaries, and testes, whereas miR-142-3p predominates in embryonic and newborn tissues [23]. Generally, the strand with the less stably paired 5′ end and a uracil or adenine at nucleotide position one is selected [37,38,39,40]. In mammals, the thermodynamic instability is sensed by AGO proteins, with only an unpaired end fitting into the AGO’s 5′ binding pocket [37,38,39]. Further, the AGO’s 5′ binding pocket has a higher affinity for the base of uracil and adenine over cytosine or guanine nucleotides [40]. Alternative Drosha processing can cause different affinities to AGO, shifting strand preference [41]. 

The mature RISC is capable of binding target mRNAs, causing mRNA instability that can reduce both target gene mRNA and protein abundance. In humans, there are four distinct AGO proteins (AGO 1–4); all four can bind microRNAs to repress target mRNA translation. However, Ago2 can also slice target mRNAs to reduce expression further.

#### 1.2.3. MicroRNA Mechanism of Action

The microRNA sequence directs RISC targeting through its “seed” nucleotides, a string of seven to eight bases near the 5′ end [42]. The supplementary region, between nucleotides 13 and 17, can stabilize target binding. RISC affinity is vital for mRNA destabilization and is established through several factors: the number of base pair matches within the seed region; additional non-seed pairing and spacing; productive 3′ pairing; and mRNA adenine and uracil content proximal to seed matching [43]. The most efficient target sites are perfect matches to all eight seed nucleotides (8mer sites), with a steep decline in affinity with seven seed matches with one neighboring non-seed match (7mer-m8), then seven seed matches and neighboring adenine (7mer-A1), followed by six seed matches (6mer), which display negligible improvement to no matches [43]. 

Additional factors beyond RISC’s affinity determine the efficiency of RISC-mediated mRNA destabilization. The location within the mRNA targeted is important: targeting the 3′ untranslated region (UTR) significantly improves RISC-mediated repression compared to targeting the 5′ UTR or open reading frame (ORF) [43]. Further, proximity to the stop codon within the 3′ UTR proximal reduces site efficacy due to competitive binding with ribosomes [43]. In addition, co-expressed microRNAs can target the same mRNA. When two microRNA-7mer target sites are next to each other within 50 nucleotides, the synergistic effect of the two 7mers can enhance their efficacy to deliver stronger suppressive results over a single 8mer [43].

All these factors should be considered when assessing potential mRNA targets. Several algorithms have been developed to predict microRNA targets that incorporate these factors [43]. Further experimental validation is necessary to demonstrate RISC-mediated mRNA repression. 

#### 1.2.4. MicroRNA Regulation in Immune Cells

MicroRNAs provide an additional layer of post-transcriptional regulation of genes. More than 60% of human protein-coding genes contain at least one conserved microRNA-target site [44]. Multiple studies have demonstrated the importance of microRNA regulation in immune functions during homeostasis and under stress [20,45]. The concept that microRNA network motifs can regulate cellular processes through both positive and negative feedback loops provides further complexity to microRNA-regulated immune cell formation and function. For example, to ensure fine-tuned cellular actions, microRNAs can establish a threshold for master regulators to a narrow range, which is crucial for tightly controlled hematopoiesis or swift immune cell response [20]. 

In this review, we will detail current knowledge on microRNA regulation in macrophage actions important to WAT functions. Multiple microRNAs have been implicated in macrophage functions, as summarized in Table 1. Although other microRNAs have been studied in macrophages, we have limited our review to data confirmed in multiple studies across independent laboratories or well-studied microRNAs implicated in relevant ATM functions. 

## 2. Adipose Tissue Changes in Obesity, Macrophage Responsibilities, and microRNA Regulation

### 2.1. Adipose Tissue Adipokines and Insulin Sensitivity

Obesity-induced chronic inflammation impairs crucial WAT metabolic regulation and exacerbates tissue and systemic insulin resistance (IR) [70,71,72,73,74]. ATMs promote tissue inflammation through pro-inflammatory cytokine expression under obese conditions. Circulating ATM-secreted cytokines can lead to systemic immune activation and, including TNF-α, can directly impair insulin sensitivity in white adipocytes and muscle [70,71,72,74]. Obese patients with WAT-IR have increased insulin-secreting beta-cell dysfunction, a precursor to Type 2 diabetes mellitus (T2D) [75]. Inflammatory cytokine neutralization improves obesity-induced systemic metabolic dysfunction [70,71,72,74].

Macrophage activation towards pro-inflammatory cytokine production is endorsed by microRNA miR-155. The first studies of miR-155 in myeloid cells by the Baltimore laboratory demonstrated miR-155 expression in monocytes [46] and macrophages [47] positively correlated with M1 activation in response to pro-inflammatory stimuli. Later studies demonstrated miR-155 promotes pro-inflammatory activation by targeting anti-inflammatory suppressor of cytokine signaling 1 (*Socs1*) [48], phosphatidylinositol-3,4,5-trisphosphate 5-phosphatase-1 (*Ship1*) [49,50], and interleukin 13 receptor subunit alpha-1 (*IL13Rα1*) [51]. MiR-155 represses expression of Socs1, a potent repressor of inflammation that inhibits several pro-inflammatory signaling cascades. Following activation by pro-inflammatory stimuli, increased miR-155 abundance controls Socs1 protein levels to allow inflammatory macrophage signaling. Ship1 is a negative regulator of kinase Akt, thus, miR-155 inhibition of Ship1 expression allows for greater Akt activity during pro-inflammatory macrophage activation [49,50]. Akt signaling is implicated in many cellular functions and triggers metabolic shifts that support macrophage activation. However, the role of Akt signaling in M1 activation is unclear. Akt signaling -increased by miR-155 may promote pro-inflammatory activation through Akt-inactivation of the TSC complex. Indeed, Tsc1-deficient macrophages produce higher levels of pro-inflammatory cytokines through the Ras GTPase-Raf1-MEK-ERK pathway [76]. IL13Rα1 is part of a cascade to induce STAT6 signaling towards anti-inflammatory activity. By suppressing IL13Rα1 expression, miR-155 prevents STAT6 activation [51]. 

Interestingly, miR-155 may exert both positive and negative regulation of pro-inflammatory signaling in macrophages at different points throughout macrophage responses [77], acting in a negative feedback loop to reduce hyper-responsiveness. However, further functional and temporal studies are required to determine the nuanced tuning of miR-155 in macrophage inflammatory signaling. It is important to note that miR-155 has been demonstrated to have more targets involved in other macrophage actions, including lipid handling [52] and anti-apoptosis [78]. In addition, beyond macrophage endogenous targets, miR-155 may be a part of ATM paracrine actions on adipocytes through exosome delivery [79,80]. Obese ATM exosomes contain abundant levels of miR-155, and their delivery into adipocytes impairs insulin-stimulated glucose uptake through decreased surface GLUT4 expression [80]. 

Other microRNAs have been described to operate in negative feedback loops to regulate macrophage activation, including miR-125b and miR-146a. MiR-125b is upregulated in alternatively-activated peritoneal macrophage in vivo (via helminth infection) and repressed in vitro by pro-inflammatory stimuli. MiR-125b directly represses *TNF-*α translation, inhibiting pro-inflammatory macrophage actions [50,53]. 

In opposition, miR-223 supports anti-inflammatory activation. MiR-223 is essential in myeloid differentiation [81,82] and is decreased during monocyte to macrophage differentiation [54]. Obese mice with systemic deletion of miR-223 had enhanced visceral AT pro-inflammatory macrophage infiltration, inflammation, and insulin resistance [55]. Activation of the transcription factor peroxisome proliferator-activated receptor-gamma (PPAR-γ) is essential for M2 macrophage activation and stimulates miR-223 expression [56]. MiR-223 promotes PPAR-γ mediated activation in a positive feedback loop by suppressing NF-κB and c-Jun N-terminal Kinase (JNK) signaling by targeting nuclear factor of activated T-cells 5 (*Nfat5*), RAS p21 protein activator 1 (*Rasa1*), and PBX/knotted 1 homeobox 1 (*PKNOX1*), resulting in reduced pro-inflammatory cytokine production [55,56,57]. Many studies have investigated microRNAs in macrophage activation across various systems. These have been reviewed elsewhere [83]. 

ATMs further perpetuate inflammation through antigen presentation to adaptive immune cells within the AT. During obesity, ATMs display increased major histocompatibility complex II (MHCII) and co-stimulatory molecules to activate CD4 + T cell proliferation and pro-inflammatory interferon-gamma (IFN-γ) production [84]. Further, ATMs have been demonstrated to display the lipid-antigen presenting molecules CD1b and CD1c [85]. Few studies have investigated microRNA regulation in the antigen presentation network, for example, [86,87], and none have used ATM-antigen presentation. 

### 2.2. Adipogenesis and Angiogenesis

Adipose tissue undergoes continual adipocyte turnover, refreshing the population through adipogenesis. Insufficient adipogenesis results in increased adipocyte size, or hypertrophy, and is correlated with increased inflammation, decreased metabolic health, and obesity-related co-morbidities [88,89,90,91,92,93]. In order to initiate adipogenesis, ATMs express osteopontin to recruit adipocyte progenitor cells near crown-like structures (CLS) [94,95]. A CLS is composed of pro-inflammatory ATMs surrounding dying adipocytes, and the localization of progenitor cells to the periphery may be a double-edged sword. Newly differentiated adipocytes have more space to expand after the CLS macrophages have cleared the dead cells [95,96]. However, TNF-α, which can be produced within the CLS, has been shown to repress adipogenesis through suppressing PPAR-γ [97,98]. In addition, specific ATMs subpopulations may present an iron repository required for adipogenesis [99,100].

MicroRNA regulation of adipogenesis within stem cells, pre-adipocytes, and mature adipocytes has been appreciated [101]; however, the intersection of ATM microRNAs on adipogenesis deserves greater investigation. Beyond endogenous control of stem cell recruitment, macrophage-derived exosome transfer of microRNAs has been demonstrated to impact various tissues and disease settings [79,80,102,103]. However, a direct network of ATM-exosomes delivered into adipogenic cells has not been investigated. 

Adipocyte differentiation is coupled with angiogenesis to deliver nutrients and growth factors as well as to prevent acidosis and hypoxia in expanding AT [104]. Increasing AT capillary density and tissue perfusion abrogates obesity-induced insulin resistance and systemic metabolic dysfunction [105]. ATMs can promote angiogenesis through direct interactions and secreted factors including matrix metalloproteinases (MMP) -7, -9, -12, vascular endothelial growth factor A (VEGF-A), fibroblast growth factor 2 (FGF2), and platelet-derived growth factor subunit B (PDGF-BB) to activate endothelial cells (ECs). Total ablation of macrophages reduces visceral AT angiogenesis [106]. A balance of pro-inflammatory and anti-inflammatory-activated macrophages is necessary to achieve de novo vessel outgrowth. However, excessive inflammation in the microenvironment prevents macrophage-supported angiogenesis [107].

Various studies have reported the regulatory effect of microRNA on angiogenesis in different diseases and are summarized elsewhere [108]. Compared to similar studies in cardiovascular diseases and cancers, ATM-produced microRNA in modulating angiogenesis has been less explored. It has been shown that exosomes from in vitro stimulated macrophages are capable of altering angiogenesis in endothelial cells [103,109]. In particular, anti-inflammatory M2 macrophage-derived exosomes promote angiogenesis in vitro and transfer functional miR-155 and miR-221 into endothelial cells [109]. Both miR-155 and miR-221 target *E2F2*, which inhibits angiogenesis in endothelial cells [109]. Although evidence supports a role for miR-221 in endothelial cells during angiogenesis [110], the importance of macrophage-derived exosome-delivered miR-155 and miR-221 deserves more research. Further, miR-155 depleted endothelial cells have been shown to undergo more robust proliferation and angiogenic tube formation in vitro [111]. In addition, miR-155 deficient macrophages exhibit defective infiltration into damaged vessels, preventing their ability to promote de novo sprout initiation [111]. This demonstrates the tissue-specific roles of microRNAs and urges further work to reveal microRNAs or targets regulating macrophage pro-adipogenic and pro-angiogenic actions independent of macrophage activation.

### 2.3. Extracellular Remodeling

The ECM of adipose tissue must be remodeled to avoid adipocyte stress via mechanosensing. Unresolved tissue remodeling and inflammation can lead to excessive ECM component deposition, known as fibrosis, which severely impairs organ functionality. Obesity-induced AT fibrosis has been linked to metabolic dysfunction resistant to weight loss [112,113,114]. ATMs can contribute to ECM clearance to allow for vascularization. However, obese ATMs’ pro-inflammatory phenotype potentiates unresolved inflammation and fibrosis [115]. In addition, macrophage transforming growth factor-beta (TGF-β) production contributes to fibroblast activation to upregulate collagen contraction and proliferation [116,117,118]. 

The miR-21 sequence is upregulated in multiple fibrotic conditions, and depletion abrogates pro-fibrogenic activity of fibroblasts to TGF-β [58,119,120,121,122,123]. In fibroblasts and tenocytes, elevated levels of miR-21 initiate their proliferation and collagen deposition. Macrophage-mediated activation of fibroblasts, via cell surface ligand-receptor interactions and secretome, is dependent on macrophage miR-21 expression; however, the mechanism is unclear [59]. One possibility is through macrophage-derived exosomes. Macrophage-derived exosomes have abundant miR-21, and co-culturing these exosomes with fibroblasts and tenocytes elevated miR-21 levels in these cells [124]. Further work is needed to elucidate miR-21 targets in macrophages that confer this action. 

In addition, miR-142 is important for myeloid progenitor generation, and its depletion impairs dendritic cell differentiation [125]. MiR-142-5p is upregulated in tissue macrophages during experimentally induced liver and lung fibrosis. Macrophage miR-142-5p knockdown decreased TGF-β production and co-cultured fibroblast collagen production [60]. Similar to miR-155, miR-142-5p represses anti-inflammatory *SOCS1* [60]. However, whether this target is involved in this action is not known. 

### 2.4. Lipid Storage and Mobilization

Lipid storage as an energy reserve is a crucial function of WAT. ATMs aid lipid storage by metabolizing extracellular free fatty acids and lipid from phagocytosed adipocytes [126,127]. ATM lipid metabolism via lysosomal lipolysis significantly reduces systemic dyslipidemia, a major driver for cardiovascular diseases [126,127]. Further, ATM lipid removal reduces inflammation by forestalling adipocyte hypertrophy and lipid-mediated immune activation. In obesity, lipid-laden ATMs are more abundant and display large lipid droplets present in bloated multinucleated macrophages [126,127,128]. 

Although all ATMs upregulate surface expression of the fatty acid transporter CD36 after a high-fat meal, a subset of ATMs delineated by phospholipid-transporting ATPase ABCA1 (Abca1), T-cell immunoglobulin and mucin domain-containing protein 4 (Tim4), and lymphatic vessel endothelial hyaluronic acid receptor 1 (Lyve1) expression are most apt for lipid uptake and metabolism [129]. These specialized ATMs are self-replicating and contribute to the cardioprotective reverse cholesterol pathway in lean and obese conditions [129]. Promoting this macrophage action is beneficial to AT functions; however, regulatory mechanisms are difficult to parse from overall ATM features. 

Research in plaque-macrophage lipid handling in atherosclerosis, termed macrophage foaming, may apply in ATM biology and has been reviewed elsewhere [130]. In particular, miR-33 targets *ABCA1* in mouse and human cells to repress cholesterol efflux [61]; anti-miR-33 treatment resulted in plaque regression via macrophage-mediated cholesterol removal [62]. In macrophages, miR-33 targets *ABCA1* and ATP-binding cassette sub-family G member 1 (*Abcg1*) to reduce cholesterol efflux [63,64]. Further, miR-33 represses human and mouse macrophage mitochondrial respiration through targeting pyruvate dehydrogenase kinase 4 (*PGC-1α*), calcium-binding mitochondrial carrier protein SCaMC-2 (*SLC25A25*), and pyruvate dehydrogenase (acetyl-transferring) kinase isozyme 4 (*PDK4*) [65]. PGC-1α deletion abrogates anti-miR-33-mediated increase in macrophage cholesterol efflux, demonstrating the importance of macrophage mitochondrial respiration in cholesterol efflux [65]. 

In addition, miR-342-5p and miR-155 are elevated in atherosclerotic lesions of Apolipoprotein E (Apoe)-knock-out mice. However, further work is needed to establish the importance of these microRNA-mRNA networks in macrophage lipid handling. MiR-342-5p targets RAC-alpha serine/threonine-protein kinase (*Akt1*) to induce IL-6 and nitric oxide synthase 2 (NOS2) production [67]. However, Akt1-deficient macrophages are not sufficient for worsened atherogenesis [131]. MiR-155 represses high-mobility group box-containing protein 1 (*Hbp1*) [52]. Hbp1 knock-out improves macrophage lipid uptake and ROS production in vitro [52], but its role in lipid handling is not clear. 

### 2.5. Age-Related Changes

Obesity-induced inflammation in adolescence and middle age accelerates the onset of declining functionality characteristic of advanced age, increasing the risk for morbidity and mortality [132]. Reciprocally, age-related disorder in AT augments obesity-induced dysfunction. Both aging and obesity are marked by metabolic dysfunction linked to chronic, low-grade inflammation. Aging defines a gradual deterioration of functionality across tissues; likewise, age-related inflammation is juxtaposed to impaired immune function. One hallmark of aging is stem cell exhaustion, best exemplified in hematopoiesis. With each division, hematopoietic stem cells (HSCs) exhibit lower self-renewal capacity and myeloid-biased differentiation [133,134]. The increased proportion of macrophages exhibits dysfunction. In aged mice, overall macrophage capacity for autophagy and phagocytosis dissipates, and ATM populations produce greater levels of pro-inflammatory cytokines [68,135].

The level of miR-146a is much higher in aged compared to young peritoneal macrophages in mice [69]. MiR-146a imposes tolerance to pro-inflammatory cytokines and danger-associated molecular patterns (DAMPs) by targeting key cytokine-receptor and TLR adaptor molecules interleukin 1 receptor-associated kinase 1 (*IRAK1*) and TNF receptor-associated factor 6 (*TRAF6*) [46,66]. Chronic overexpression of miR-146a prevented M1 effector activation (IL-1β, IL-6, TNF-α production) in response to bacterial lipopolysaccharide (LPS) stimulation in vitro [69]. In addition, peritoneal macrophage miR-33 increases in aged mice, leading to reduced cholesterol efflux [136]. MicroRNA regulation in macrophage actions under chronic conditions like obesity and aging-related disorder has yielded valuable insight into epigenetic factors controlling disease risk.

## 3. How to Capture Dynamic Actions of Macrophages in Obese Adipose Tissue

### 3.1. Current Views

Traditionally, macrophages are characterized by their activation status, which can be described along a spectrum from the classical M1 phenotype and the alternatively activated M2 phenotype [137]. M1 macrophages provide acute pro-inflammatory effector functions by expressing reactive oxygen species (e.g., hydrogen peroxide), nitric oxide, and secretion of type-1 cytokines such as TNF-α, IL-1β, and IFN-γ. LPS and IFN-γ stimulate M1 activation through activation of the Janus kinase-signal transducer and activator of transcription (JAK-STAT) pathway via NF-κB and mitogen-activated protein kinases (MAPK) signaling [138]. LPS binds to TLR4 on cell surfaces to induce downstream signaling. Further, interferon regulatory factors (IRFs) modulate IFN secretion. M1 activation is promoted by IRF-1, -2, -5, and -6. In contrast, IRF-3 and -4 mediate anti-inflammatory signaling and M2 activation [138]. 

M2 actions include resolving acute inflammation and secreting type 2 cytokines like IL-10 and arginase-1. Polarization to alternative M2 activation is achieved by IL-4 and IL-13 stimulation; M2 activation depends on the transcription factor PPAR-γ. STAT6 is associated with M2 activation as it mediates IL-4 signaling; however, prolonged phosphorylation is associated with pro-inflammatory features [138]. STAT6 may also act as a co-factor to PPAR-γ, promoting its transcriptional capacity. The C/EBP pathway mediates M2 arginase-1 production and inhibits M1-associated gene expression, despite being a target of PPAR-γ-induced miR-223 [138]. 

However, macrophages respond to a plethora of stimuli beyond LPS/IFN-γ versus IL-4/IL-13 activation. In particular, ATMs are exposed to free fatty acids, and other extracellular lipids have been implicated in TLR4 signaling characteristic of M1 activation [139]. Regardless, ATMs do not display surface markers characteristic of M1 or M2, and various distinct subpopulations persist [140]. MicroRNAs and other epigenetic regulators shape these unique macrophage features to exert necessary functions towards AT homeostasis. 

### 3.2. Investigation Strategy 

MicroRNAs’ regulatory power and potential as disease biomarkers have motivated rapid advancement in microRNA detection and experimentation. A review of strategies for microRNA detection can be found elsewhere [141]. The M1/M2 model of macrophage activation is an essential tool for in vitro validation of microRNA regulation of macrophage activation; however, more complex models are needed to understand ATMs and other in vivo macrophage functions. Investigations using the M1/M2 model have defined miR-155 and miR-223 as crucial regulators of macrophage activation towards pro-inflammatory or anti-inflammatory actions. 

### 3.3. Incorporating Macrophages Diversity in Studies

ATMs are not easily classified as M1/M2 activated. During an inflammatory response in vivo, macrophages experience diverse, multilayered signals, resulting in heterogeneous populations of activation and functionality. Thus, researchers have made efforts towards analyses that employ complex models that consider macrophage activation along a continuum and to varying stimuli and functional studies to understand ATMs [95,140,142,143].

To this end, we have developed two transcriptome-based tools with single-cell resolution to annotate monocyte/macrophages along action axes: MacSpectrum [144] and AtheroSpectrum [145]. These tools depict transitional states towards macrophage features, allowing for in-depth investigation into regulatory mechanisms involved in promoting macrophage programs. MacSpectrum depicts monocyte and macrophage differentiation and pro-inflammatory/anti-inflammatory activation [144]. Application of MacSpectrum to single-cell transcriptomics of lean and obese ATMs recapitulated the increase of pro-inflammatory macrophages characteristic of obesity [144]. AtheroSpectrum likewise quantifies pro-inflammatory/anti-inflammatory activation in addition to macrophage-derived foam cell differentiation important to atherosclerosis [145]. AtheroSpectrum revealed two distinct foaming programs that occur in mouse and human plaques: homeostatic foaming and pathogenic foaming [145]. Homeostatic foaming is anti-inflammatory and present in health and atherogenic plaques, in line with macrophages’ continual role in maintaining vessel integrity. Pathogenic foaming was characterized by inflammatory foaming and correlated with cardiovascular disease (CVD) incidence and severity. Identification of these novel foaming programs allowed for focused investigation into pathogenic foaming, prompting the development of a CVD risk assessment that incorporates a pathogenic-foaming gene-set [145]. Models and tools to depict macrophage plasticity are necessary to parse out nuanced regulatory networks driving diversified macrophage function, including microRNA dynamics.

## 4. Summary

During obesity, essential AT metabolic regulation and lipid handling are disrupted due to changes in adipokine production, insulin sensitivity, adipogenesis, angiogenesis, and extracellular remodeling. ATMs potentiate chronic AT inflammation leading to systemic metabolic dysfunction. Importantly, ATMs are responsible for heterogeneous functions in obese and lean conditions, responding to the stimuli within their microenvironment to modulate AT homeostasis. MicroRNAs tune macrophage intracellular signaling and activation, promoting diversified function within the AT and throughout the body. Figure 1 depicts ATM functions in obesity and microRNA thus far revealed to regulate these macrophage actions, including extracellular remodeling, adipocyte clearance, promotion of adipogenesis and angiogenesis, and lipid uptake. Additional ATM functions have been observed in lean and obese conditions, including antigen presentation for T cell activation and iron sequestering. ATM heterogeneity makes applying the popular M1/M2 model difficult and urges investigators to apply complex models and tools that consider the continuum of macrophage activation and function. ATMs exert specific functions within distinct niches across AT. Utilization of emerging spatial transcriptomics with spatial proteomic technologies will depict the ATM mosaic landscape and provide insight into transcriptional programs driving niche-specific ATM functions.

## Figures and Tables

**Figure 1 cells-11-01336-f001:**
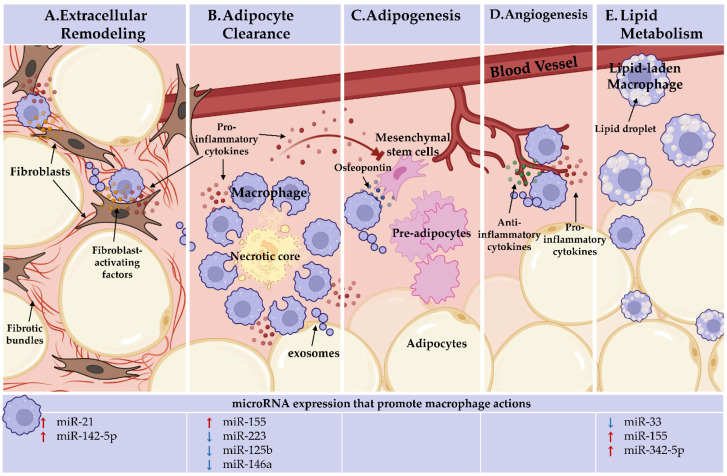
Adipose tissue macrophages (ATMs) perform heterogeneous, niche-specific functions and under obese stress perpetuate pathologic inflammation. In the top panel, ATM functions. In the bottom panel, the microRNAs known to endorse (red arrows) or inhibit (blue arrows) these macrophage actions are detailed. (**A**) Extracellular remodeling: ATMs support AT extracellular matrix remodeling. ATMs can activate fibroblasts through direct interactions and paracrine signaling, licensed by miR-21 and miR-142-5p, to increase tissue fibrosis under obesity. (**B**). Adipocyte clearance: pro-inflammatory ATMs clear dead and dying adipocytes in crown-like structures (CLS) and miR-155 supports, whereas miR-223, miR-125b, and miR-146a repress pro-inflammatory macrophage activation. (**C**) Adipogenesis: ATMs recruit progenitor cells through osteopontin secretion. Inflammatory ATM cytokine signaling within CLS, which miR-155, miR-223, miR-125b, and miR-146a regulate, impacts progenitors. ATMs may deliver microRNAs through exosomes to regulate adipogenesis. No endogenous role for microRNAs in ATM promotion of adipogenesis has been identified. (**D**) Angiogenesis: pro-inflammatory and anti-inflammatory-activated macrophages are required for de novo vessel outgrowth. Similar to adipogenesis, ATM may regulate angiogenesis through microRNA-containing exosomes. Beyond the importance of macrophage activation status, no endogenous role for microRNAs in ATM promotion of angiogenesis has been identified. (**E**) Lipid metabolism: ATMs uptake and metabolize lipids. Macrophage lipid efflux is repressed by miR-33. MiR-155 and miR-342-59 may play a role as they are upregulated during fatty plaque formation in vessels (atherosclerosis), but their networks towards lipid efflux has not been defined. Figure Created with BioRender.com (accessed on 13 February 2022).

**Table 1 cells-11-01336-t001:** MicroRNAs and confirmed mRNA targets that regulate macrophage functions important to AT function.

MicroRNA	Effects	Confirmed mRNA Targets	Source
miR-155	Pro-inflammatory activation	*Socs1*, *Ship1*, *IL7Rα1*	[46,47,48,49,50,51]
	Pro-foam cell	*Hbp1*	[52]
miR-125b	Anti-inflammatory activation	*TNF-α*	[50,53]
miR-223	Anti-inflammatory activation	*Nfat5*, *Rasa1*, *PKNOX1*	[54,55,56,57]
miR-21	Pro-fibroblast-activating interactions		[58,59]
miR-142-5p	Pro-fibroblast-activating interactions	*SOCS1*	[60]
miR-33	Anti-cholesterol efflux/ Metabolic reprogramming	*ABCA1/Abcg1/PGC-1α/PDK4/SLC25A25*	[61,62,63,64,65,66]
miR-342-5p	Pro-foam cell	*Akt1*	[67]
miR-146a	Anti-inflammatory activation	*IRAK1, TRAF6*	[46,68,69]

## Data Availability

Not applicable.

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
