# Peer review of "Impact of microRNA Regulated Macrophage Actions on Adipose Tissue Function in Obesity"

_cells, 2022, doi:10.3390/cells11081336_

Round 1

Reviewer 1 Report

The paper entitled "Impact of microRNA regulated macrophage actions on Adipose Tissue Function in obesity" and submitted for publication by the group of Dr B. Zhou, realizes an interesting review about the cells involved in inflammation of the fatty tissue such as macrophages as well as the role of microRNAs and their functions on macrophages of adipose tissue.
The presentation of the review is correct, however some changes in it could improve the quality of the manuscript.
The paragraph "1.3 MicroRNA biogenesis, mechanism of action, and regulation of macrophage function" should be written more clearly. Each of the sections Biogenesis, mechanism of action... should be titled separately and include a diagram or figure to facilitate reading. In this format is dense.
I consider that it should be clarified in the text to which type of adipose tissue the authors is referring. It is specified that it is white adipose tissue, but nothing is said about whether the studies to which it refers have been carried out in subcutaneous, visceral, or omental adipose tissue. The contribution of macrophages in the inflammation in these types of tissues is different and their participation in inflammation is also different, therefore they should be referred to.
The bibliography is very complete and up-to-date. 
In line 385, foot of figure 1, there is an erratum "C "angiogenesis" it has to say “adipogenesis”

Reviewer 2 Report

Good reviews explain difficult topics and make use of examples to illustrate how phenomena or theories are connected, or provide synthetic overviews of a large body of literature (empirical or theoretical). In this manuscript, the authors claimed that adipose tissue macrophages (ATM) functions in the context of obesity-induced adipose tissue (AT) abnormalities and the microRNAs that license macrophage roles in AT are discussed, while they just mainly talked about the different functions of miR-155, miR-223, and miR-125b which are shown in Figure 1. The authors devoted large contents regarding ATM affecting AT, however, this is nothing new. In my opinion, a more critical and mechanistic appraisal of this field is required. This manuscript consists of a large amount of foreshadowing knowledge, could be synthesized and condensed, and could be more logical. Also, it seems that the authors didn’t discuss the underlying mechanisms that how these microRNAs impact on macrophage actions. What are the downstream signaling pathways which are altered? Some more attention could be paid to this aspect. And the article will need a careful appraisal for English grammar. In particular, the tense is often inconsistent (changing between past and present tense). An example is in the first paragraph of section 1.2. 

Reviewer 3 Report

Obesity is a pandemic disorder that is distinguished by the accumulation of adipose tissue and chronic low-grade inflammation that is caused primarily by adipose tissue macrophages (ATMs). In this context, miRNAs important to immune function and potential as inflammation-modulatory targets have been highlighted by numerous studies. Thus, the authors discuss ATM functions in the context of obesity-induced AT abnormalities and the impact of the miRNAs on the macrophage roles in AT. The topic is original and relevant in the field and addresses a specific gap in the field. I believe this review would be very useful for the clinical perspective in Obesity-induced adipose tissue (AT) dysfunction/miRNAs axis. Additionally, this review provides new insight into Obesity-induced adipose tissue (AT) dysfunction clinical practice. The references are well updated. I found the conclusion to be in line with the evidence and arguments presented, and yes, the authors address the main question beautifully. 

There is a single minor concern that authors should address. The table is okay but the Figure is not well explained. For the readers, I found it would be difficult to understand.  I didn't find inside figure A. Extracellular remodeling? and B? and C? and D? and E?. What is the 
red arrow in the figure?  What are the red/green/orange dots?

Round 2

Reviewer 2 Report

Thanks for the contents added by the authors, and they have addressed some of my concerns. The authors claimed that they have limited this review to data confirmed in multiple studies across independent laboratories or well-studied microRNAs implicated in relevant adipose tissue macrophages functions, nevertheless, only eight microRNAs and five types of physiological roles have been mentioned in this article, and microRNAs appear to be involved in only three functions of adipose tissue macrophages as shown in Figure 1. Therefore, the evidence provided are obviously not enough to support the proposed big idea or concept that the regulatory networks governed by microRNAs in ATMs play an important role in white adipose tissue.

Author Response

We have read your comments carefully and evaluated the merit of this

submission, which was appreciated by Reviewer 1 and Reviewer 3.

We believe, our review on "How microRNA regulated macrophage actions on adipose tissue" focuses on providing readers a summary of current known knowledge on this topic (that we chose credible and meaningful reports to be included) in an organic structure. As you pointed out that was also appreciated by R1 and R2 that we have covered up-to-date information on all five primary functions of adipose tissue, and summarized key discoveries on how microRNAs orchestrate macrophage  actions and contribute to adipose tissue stress-responding changes.

We respectfully disagree with Reviewer 2's comments about merely increasing the inventory list of miRs.